# Lamin A/C Is Dispensable to Mechanical Repression of Adipogenesis

**DOI:** 10.3390/ijms22126580

**Published:** 2021-06-19

**Authors:** Matthew Goelzer, Amel Dudakovic, Melis Olcum, Buer Sen, Engin Ozcivici, Janet Rubin, Andre J. van Wijnen, Gunes Uzer

**Affiliations:** 1Department of Mechanical & Biomedical Engineering, Boise State University, 1910 University Drive, MS-2085, Boise, ID 83725, USA; matthewgoelzer@u.boisestate.edu; 2Mayo Clinic, Rochester, MI 55905, USA; dudakovic.amel@mayo.edu (A.D.); vanwijnen.andre@mayo.edu (A.J.v.W.); 3Department of Medicine, University of North Carolina, Chapel Hill, NC 27599, USA; melis@email.unc.edu (M.O.); buer_sen@med.unc.edu (B.S.); janet_rubin@med.unc.edu (J.R.); 4Department of Bioengineering Izmir Institute of Technology, Urla 35430, Turkey; enginozcivici@iyte.edu.tr

**Keywords:** lamin A/C, LINC, nucleoskeleton, nuclear envelope, adipogenesis, mechanical signals, mesenchymal stem cells

## Abstract

Mesenchymal stem cells (MSCs) maintain the musculoskeletal system by differentiating into multiple lineages, including osteoblasts and adipocytes. Mechanical signals, including strain and low-intensity vibration (LIV), are important regulators of MSC differentiation via control exerted through the cell structure. Lamin A/C is a protein vital to the nuclear architecture that supports chromatin organization and differentiation and contributes to the mechanical integrity of the nucleus. We investigated whether lamin A/C and mechanoresponsiveness are functionally coupled during adipogenesis in MSCs. siRNA depletion of lamin A/C increased the nuclear area, height, and volume and decreased the circularity and stiffness. Lamin A/C depletion significantly decreased markers of adipogenesis (adiponectin, cellular lipid content) as did LIV treatment despite depletion of lamin A/C. Phosphorylation of focal adhesions in response to mechanical challenge was also preserved during loss of lamin A/C. RNA-seq showed no major adipogenic transcriptome changes resulting from LIV treatment, suggesting that LIV regulation of adipogenesis may not occur at the transcriptional level. We observed that during both lamin A/C depletion and LIV, interferon signaling was downregulated, suggesting potentially shared regulatory mechanism elements that could regulate protein translation. We conclude that the mechanoregulation of adipogenesis and the mechanical activation of focal adhesions function independently from those of lamin A/C.

## 1. Introduction

As one of the lamin family proteins that form the nucleoskeleton, lamin A/C (gene symbol: *Lmna*) plays a vital role in providing the mechanical and structural integrity of the cell nucleus [1,2,3]. Mutations in *Lmna* lead to premature aging in Hutchinson–Gilford progeria syndrome [2,3,4], also known as progeria [1,2,3], where the progeroid farnesylated lamin A/C (i.e., progerin) causes alterations to heterochromatin and DNA damage [3,4,5,6]. Mechanical and structural attributes of the cell and nucleus also change during loss and mutation of lamin A/C [7,8]. Nuclei containing progerin show increased stiffness when visualized under strain [9]. When lamin A/C is depleted, cellular elasticity and viscosity of the cytoplasm decrease [10]. Additionally, irregular nuclear morphology, including blebbing and loss of circularity, is attributed to progerin presence and lamin A/C depletion [7,8]. Such a change in mechanical properties affects the response to external forces. The nucleus of lamin-A/C-deficient cells displays a higher displacement magnitude in response to biaxial strain, indicating lower nuclear stiffness compared to wild-type nuclei [11]. Therefore, lamin A/C plays a critical role in determining human health and regulating cellular and nuclear mechanical structure and shape.

Mesenchymal stem/stromal cells (MSCs) are tissue-resident somatic multipotent cells that can differentiate into musculoskeletal lineages, including osteoblasts and adipocytes [12]. Depletion of lamin A/C in MSCs impedes osteoblast differentiation. MSCs treated with an siRNA targeting *Lmna* shows reduced osteoblast differentiation transcription factors such as osteocalcin (*Bglap*), osterix (*Sp7*), and bone sialoprotein (*Ibsp*) and an increase in fat droplet formation [13]. Additionally, mouse studies have shown that *Lmna*^−/−^ mice have a significant reduction in bone mass compared to WT mice, reflecting reduced osteoblast numbers [14]. Lamin A/C overexpression induces osteogenesis while simultaneously inhibiting adipogenesis in human MSCs [15]. In addition to causing progeria and inhibition of osteogenesis, mutations of *Lmna* also produce lipodystrophy syndromes. Heterozygous mutations of *Lmna* cause familial partial lipodystrophy, Dunnigan variety (FPLD2), which is characterized by the loss of subcutaneous fat in the upper and lower extremities [16]. The FPLD2-associated *Lmna* p.R482W mutation slows adipogenic differentiation in fibroblastic cells extracted from FPLD2 patients and in human adipose stem cells [17]. At the same time, lamin-A/C-depleted MSCs, when cultured on softer substrates, are more permissive of adipogenic differentiation compared to control MSCs [18]. While these findings suggest a role for lamin A/C in regulating differentiation at both the organism and the cellular level, whether lamin A/C depletion contributes to mechanical regulation of MSC differentiation remains insufficiently explored.

MSCs replace and rejuvenate skeletal and connective tissues in response to environmental mechanical signals [19,20,21]. For example, application of external mechanical challenge in the form of LIV over 14 days increases proliferation and osteogenic differentiation markers and subsequent mineralization of MSC cultures in vitro [22,23]. An important signaling node for mechanical control of MSCs is focal adhesions, macromolecule protein complexes located on the cellular membrane, that connect the cytoskeleton to the extracellular matrix (ECM) where the cell is anchored to the extracellular environment through integrins [24]. During dynamic mechanical stimulus, integrin engagement is regulated by activation of focal adhesion kinase (FAK) at the tyrosine 397 residue [25]. We have reported that both LIV and substrate strain lead to FAK phosphorylation at tyrosine 397 [26]. This activation of FAK at focal adhesions both recruits signaling molecules that lead to cytoskeletal restructuring and activates concomitant mechanosignaling events such as the Akt/ß-catenin (*Akt1*/*Ctnnb1*) pathway [27]. For example, application of mechanical stimuli with strain, fluid flow, and LIV has been shown to activate both ß-catenin and RhoA signaling in MSCs [26,28,29]. Within the context of MSC adipogenesis, activation of these parallel signaling pathways results in decelerated adipogenic commitment of MSCs, as measured by reduced production of adipogenesis-related proteins such as adiponectin (encoded by the adiponectin gene *Adipoq*) and peroxisome-proliferator-activated receptor gamma (*Ppparg*1) [30].

In addition to cytomechanical signaling events initiated at focal adhesions and the cytoskeleton, control of MSC differentiation is also dependent on nuclear connectivity within the cytoskeleton. Inhibiting nucleo-cytoskeletal connectivity by disabling the function of linker of nucleoskeleton and cytoskeleton (LINC) complexes impedes the nuclear entry of important molecular transducer mechanical information such as YAP/TAZ and ß-catenin, which act as co-transcriptional factors for regulating MSC adipogenesis and osteogenesis [31]. As opposed to LINC complex depletion, lamin A/C depletion has no effect on mechanically induced nuclear ß-catenin entry [32], suggesting that lamin A/C may be dispensable for the mechanically induced activation of focal adhesions that lead to the de-phosphorylation and subsequent nuclear entry of ß-catenin. These previous studies show that lamin A/C plays a central role in nuclear organization and structure, as well as contributing to the cell’s ability to sense structural qualities of the extracellular matrix to guide differentiation of MSCs. However, the role of lamin A/C in focal adhesion signaling and mechanically induced control of MSCs’ fate in response to dynamic mechanical challenges remains incompletely understood. Here we tested the requirement of lamin A/C for the mechanical response of MSCs. Using LIV, we investigated the role of lamin A/C depletion on the mechanical control of MSC adipogenesis.

## 2. Results

### 2.1. siRNA Depletion of Lamin A/C Weakens the Nuclear Elastic Modulus in MSCs

We investigated the effects of lamin A/C depletion on cellular and nuclear morphology as well as mechanical properties. MSCs treated with either a control siRNA (siCntl) or a lamin-A/C-specific siRNA (siLmna) were stained against F-actin and DNA. Compared to the siCntl group, siLmna-treated MSCs showed a more elongated nuclear morphology but no apparent changes in the F-actin cytoskeleton (Figure 1a). As shown in Figure 1b, 3D morphology quantification indicated a 9% decrease in nuclear sphericity in siLmna-treated MSCs when compared to MSCs treated with a control siRNA (*p* < 0.001). The maximal nuclear area, volume, and height were increased by 32%, 31%, and 11% in siLmna-treated MSCs, respectively, when compared to siCntl-treated cells (Figure 1b; *p* < 0.001). Young’s modulus was measured for both whole cells and extracted nuclei treated with either siLmna or siCntl. Young’s modulus was measured using a rounded AFM probe tip, which was pressed onto the surface of the whole cell directly above the nucleus or on an isolated nucleus (Figure 1c,d). Confocal imaging with DNA and lamin A/C labeling of a representative isolated nucleus (Figure 1e) indicated that the nuclear structure remains intact following isolation. Treatment with siLmna caused a 45% reduction in whole-cell stiffness when compared to siCntl-treated MSCs (Figure 1f; *p* < 0.001), while extracted nuclei exhibit a 55% reduction in stiffness in lamin-A/C-depleted cells compared to the siCntl group (*p* < 0.01, Figure 1g).

### 2.2. siRNA Depletion of Lamin A/C Increases SUN2 Nuclear Levels and Focal Adhesion Proteins

To further characterize the effects of lamin A/C depletion on the nuclear envelope and focal adhesions, the LINC complex and focal adhesion proteins were investigated. Confocal images of the siCntl and siLmna groups indicated that there were no visible changes in the LINC proteins SUN1 and SUN2 when lamin A/C was depleted (Figure 2a). Quantitative analysis of the confocal images detected no differences in SUN1 or SUN2 nuclear envelope localization (Appendix A Appendix A). Hoechst-stained images can be found in Appendix A Appendix A. We examined the same proteins using cellular fractionation followed by Western blotting and densitometry analysis (Figure 2b,c). All the measurements were normalized to whole-cell siCntl protein amounts, which were set to 1. Comparing siLmna treatment with siCntl, lamin A/C was significantly decreased in whole-cell fractions (−35%, *p* < 0.05). The relative lamin A/C concentration was greater in the nuclear fraction and led to larger values, while band intensities of the siLmna group remained significantly lower compared to the siCntl group (−11%, *p* < 0.05). Except for a small amount of SUN1 detection in the cytoplasm, both SUN1 and SUN2 were largely restricted to the nucleus. In contrast to the results from confocal imaging (Appendix A Appendix A), Western blots quantifying cell fractionations from lamin-A/C-depleted cells showed an increase in nuclear SUN2 (+44%, *p* < 0.05). Focal adhesion proteins were also altered under siLmna treatment. Total focal adhesion kinase (FAK) adhered to the cell culture plate experienced an increase of 39% compared to control-treated cells (*p* < 0.05; Figure 2d,e). The amount of Akt adhering to cell culture plates also increased by 50% (*p* < 0.05). No changes in vinculin were detected.

### 2.3. Focal Adhesions Maintain Response to Mechanical Stimulus in Lamin-A/C-Depleted MSCs

Based on the observed increases in basal levels of FAK during lamin A/C depletion (Figure 2d,e) we next asked whether mechanical activation of FAK is altered by further quantifying the mechanical activation of FAK via its phosphorylation at the tyrosine 397 residue (pFAK), which is indicative of integrin engagement [25]. MSCs were treated with either strain or LIV and compared to non-mechanically stimulated controls. Basal pFAK levels normalized to total FAK (TFAK) were 85% elevated in the siLmna group when compared to the siCntl group (*p* < 0.05) (Figure 3a,b). Phosphorylated FAK levels in both siCntl- and siLmna-treated groups increased by 101% (*p* < 0.05) and 87% (*p* < 0.001), respectively, in response to a 20 min strain (2%, 0.1 Hz) when compared to non-strained counterparts (Figure 3c,d). LIV also activated FAK, increasing pFAK by 331% (*p* < 0.001) in siCntl- and 83% (*p* < 0.001) in siLmna-treated MSCs in response to LIV (0.7 g, 90 Hz) (Figure 3a,b).

### 2.4. Application of Daily LIV Treatment Decreases Adipogenic Differentiation in MSCs

As focal adhesion signaling was intact in siLmna-treated MSCs, we next probed downstream processes to ask whether LIV application, known to slow adipogenesis [33], was effective when lamin A/C was depleted. In our experiment timeline, cells were first treated with siRNA on day 1 and then cultured in adipogenic media concomitant with LIV treatment up to 7 days (Figure 4a). On day 2, adipogenic media were placed on cells and LIV treatment started. LIV treatment occurred twice a day for 20 min with 2 h rests in between treatments. On day 7, cell protein, imaging plates, and RNA samples were collected together for Western blotting, lipid droplet staining, and RNA-seq analysis. Probing the adipogenesis marker adiponectin (gene symbol: *Adipoq*) between non-LIV controls, lamin-A/C-depleted cells showed a 39% decrease in adiponectin at 7 days (Figure 4c,d). Similarly, compared within LIV-treated groups, adiponectin levels in the siLmna group was 51% lower than in siCntl-treated cells with LIV (*p* < 0.01). Compared to non-LIV controls, daily LIV application decreased adiponectin levels by 30% in the siCntl group (*p* < 0.01) and 44% in the siLmna group (*p* < 0.001). To further validate the reduction in adipogenesis during LIV treatment, lipid droplets were imaged via Oil Red O (Figure 4d) and LipidSpot 610 florescent lipid droplet staining (Biotium, CA; Figure 4e). Quantification of the mean florescent lipid droplet intensity per cell from individual imaging fields revealed that LIV-treated siCntl samples had a reduction of 49% in lipid droplet intensity when compared to non-LIV-treated siCntl samples (*n* = 36, *p* < 0.0001) (Figure 4f). Non-LIV-treated siLmna samples had a 42% decrease in lipid droplet intensity compared to non-LIV-treated siCntl samples (*n* = 36, *p* < 0.001). When comparing non-LIV- and LIV-treated siLmna samples, lipid droplet intensity per cell was reduced in LIV-treated samples by 44% (*n* = 36, *p* < 0.05).

### 2.5. Differential Effect of Lamin A/C Depletion and LIV on mRNA Transcription during Adipogenic Differentiation

RNA-seq was performed to determine the effects of LIV and siLmna treatment on differential expression of mRNA in MSCs during adipogenesis. Read values were filtered for expression by selecting genes with average minimal levels of 0.3 fragments per kilobase of transcript per million mapped reads (FPKM; *t*-test *p* < 0.05 and fold change > 1.4). Hierarchical clustering of these genes generated a heatmap (Figure 5a) in which siCntl-treated samples clustered together in one clade, while undifferentiated and siLmna-treated samples were clustered together in another clade that was visually separated from siCntl-treated samples. Principal component analysis (Figure 5b) showed further grouping of siCntl and siLmna samples. Principal components 1 and 2 explained 40.4% and 15.9% of the total variance, respectively, with prediction ellipses indicating the probability of 0.95 that a new observation of the same group will fall inside the ellipse. FPKM levels for adipogenic genes with at least one group having significantly differentially expression (log_2_ fold change ≥ 1, Wald *p* < 0.05) compared to the non-LIV siCntl group as detected by DESEQ2 analysis (Figure 5c). Results showed significant differential expression between siCntl and siLmna groups.

### 2.6. Lamin A/C Depletion Impedes Adipogenic Transcription in MSCs

DESEQ2 analysis was conducted on non-LIV siLmna and siCntl groups to determine differential gene expression between siRNA treatments during adipogenic treatment. A volcano plot for the comparison between siLmna- and siCntl-treated samples under adipogenic constraints (Figure 6a) revealed that there are 52,607 statistically unchanged transcripts between lamin-A/C-depleted and control MSCs with Wald values of *p* > 0.05 (gray and green data points). As shown in green data points, 2000 genes showed at least a 2-fold difference (i.e., log_2_ fold change ≥ 1), while 749 genes showed a statistically significant change between lamin-A/C-depleted and control MSCs with Wald values of *p* < 0.05 (shown in blue), and 427 of them had a less than 2-fold difference (i.e., log_2_ fold change ≤ 1). The remaining 322 genes showed at least a 2-fold difference (i.e., log_2_ fold change ≥ 1), which represents significant and differentially expressed genes. Upregulated (red genes on the right side, *n* = 173) and downregulated genes (red genes on the left side, *n* = 149) upon lamin A/C depletion were then assessed by clustering analysis using STRING [34]. As highlighted in Figure 6b, upregulated genes upon lamin A/C depletion were associated with cellular processes such as (i) tissue repair (e.g., genes generally involved in angiogenesis, hematopoiesis, and mechanical stress shielding), (ii) ECM remodeling (e.g., genes generally involved in take-up and intra-cellular transport of ECM debris as well as suppression of apoptosis), and (iii) cell surface transporters (e.g., genes that mediate the trafficking of compounds across membranes. The detected GO term pathways with FDR < 0.05 are presented in Appendix A Appendix A. Collectively, the biological function of these genes appears to be related to tissue repair, inflammation, and extracellular matrix homeostasis. As depicted in Figure 6c, downregulated gene groups upon lamin A/C depletion included (i) cell adhesion and cytoskeletal organization, (ii) interferon signaling and regulation of gene expression (e.g., DNA and RNA binding and protein degradation), (iii) G-protein-coupled receptor signaling (e.g., diverse range of cell surface receptors and components of the angiotension system), (iv) lipid metabolism and paracrine inflammatory signaling, and (v) adipogenic phenotype. The detected GO term pathways with FDR < 0.05 are presented in Appendix A Appendix A. These downregulated genes together are generally involved in cell migration, energy metabolism, and adipogenic differentiation. The results from Gene Ontology and gene network analysis revealed that lamin A/C depletion has pleiotropic effects on gene expression, yet many gene pathways converge on cell-surface-related biochemical events, interactions with the extracellular matrix, and internal metabolic pathways.

### 2.7. Low-Intensity Vibration (LIV) Decreases Interferon Signaling Pathway Gene Expression

To determine the effects of LIV, DESEQ2 analysis was performed comparing non-LIV siCntl and siLmna controls against their LIV-treated counterparts during adipogenesis. The volcano plot comparing siCntl adipogenesis with or without LIV treatment (siCntl ± LIV) is shown in Figure 7a. There were 53,326 statistically unchanged genes between with Wald values of *p* > 0.05 (gray and green data points), with 1939 of them showing at least a 2-fold difference (green). While 76 genes showed a statistically significant change between LIV-treated and control MSCs with Wald values of *p* < 0.05, shown in blue, 26 of them had a less than 2-fold difference. The remaining 53 genes showed at least a 2-fold difference (red). Assessing downregulated genes via clustering revealed an interferon-related gene cluster in the LIV treatment group (Figure 7b). Similarly, LIV treatment upon lamin A/C depletion also revealed an interferon-related gene cluster that was downregulated (Figure 7d). Together, cells treated with siCntl had 11 genes and those treated with siLmna had 16 genes associated with the interferon signaling pathway that experienced decreased expression during LIV treatment (Figure 7e). Excluding this latter finding, it appears that while lamin A/C loss has a dramatic impact on gene expression programs, LIV has minimal effects on the transcriptome of differentiating MSCs.

## 3. Discussion and Conclusions

In lamin-A/C-depleted cells, microscopic observations of increased blebbing, elongated nuclear shape, and ruffled nuclear membrane [7,35,36] indicate a compromised nuclear structure. Our quantification of the 3D nuclear structure of lamin-A/C-depleted cells supported these previous observations and showed reduced sphericity and an increased planar nuclear area, while nuclear height and volume increased compared to controls. It has been reported that lamin A/C depletion increases nuclear height and volume in part due to reduced recruitment of perinuclear and apical F-actin cables [37]. While reduction in apical F-actin may contribute to a decrease in elastic modulus in lamin-A/C-depleted intact MSCs, a similar decrease was observed in our lamin-A/C-depleted isolated live nuclei. The similarities in decreased stiffness in both intact cells and isolated nuclei suggest that nuclear softening is the primary driver of decreased cell stiffness upon lamin A/C loss of function and that loss of lamin A/C is a major contributor to nuclei stiffness.

Our data suggest that MSCs compensate for lamin-A/C-mediated nuclear softening by increasing their focal adhesions. Not only was total FAK and Akt accumulation at focal adhesions more robust in lamin-A/C-depleted MSCs, but also tyrosine 397 phosphorylated FAK (pFAK) was higher, which suggests increased integrin engagement [25]. These findings are not surprising as depletion of both lamin A/C [38] and nucleo-cytoskeletal connector nesprin-1 (*Syne1*) [39] are shown to increase substrate traction in cells. Both LIV and strain pushed acute FAK phosphorylation of lamin-A/C-depleted cells higher than control cells, indicating that the focal adhesion signaling remains intact despite lamin A/C depletion in MSCs. Phosphorylation of FAK in siCntl and siLmna cells during LIV increased proportionally to their basal pFAK levels. However, strain-induced pFAK increases were increased to similar levels of LIV-treated siLmna samples despite being a stronger mechanical force. It is known that the number of focal adhesions determine the amount of pFAK [21,40]. While it is possible that LIV and strain activate pFAK differently, our previous findings show that LIV and strain responses are additive [26], and therefore this option is unlikely. Therefore, it possible that the strain-activated pFAK was saturated and did not increase pFAK levels higher than that of LIV mechanical activation. Ultimately, these data show that lamin A/C is also not required for the activation of focal adhesions during mechanical stimulation.

Similar to focal adhesions, nucleo-cytoskeletal connectivity provided by the LINC complex remained intact under lamin A/C depletion. Previous studies have shown that LINC proteins SUN1 and SUN2 bind to the lamin A/C in order to mediate a connection from the inner nucleus to the cytoskeleton and ultimately to the focal adhesions that make a physical connection to the extracellular matrix [41,42]. Quantification of confocal images of SUN1 and SUN2 revealed no changes compared to controls. However, quantification of three biological replicates showed that nuclear SUN2 protein levels were increased. We previously reported that depletion of LINC complex function impedes LIV-mediated FAK activation in MSCs [26]. As shown in Figure 3a, basal FAK phosphorylation levels were higher, and LIV resulted in a larger FAK phosphorylation response of lamin-A/C-depleted cells, indicating that the LINC complex function is intact despite lamin A/C depletion. This is not surprising as the localization of SUN1 and SUN2 proteins to the nuclear envelope is not entirely dependent upon lamin A/C [42,43]. While noted changes in SUN proteins after lamin A/C depletion suggest a putative relationship (Figure 2c), loss of lamin A/C did not negatively impact the SUN protein levels, suggesting that LINC complex connectivity remains intact.

Adipogenesis has recently been shown to decrease with mutated *Lmna*, specifically in cells expressing the lipodystrophy-associated *Lmna* p.R482W mutation [17]. Our data support this previous observation: MSCs treated with siLmna experienced slower adipogenic differentiation compared to siCntl-treated cells, as marked by reduced adiponectin levels and reduced lipid droplet formation and intensity (Figure 4c,f). Further investigation revealed that siLmna-treated MSCs show a reduction in lipid droplet intensity compared to siCntl-treated MSCs (Appendix A Appendix A). A multiple linear regression analysis with two- and three-way interactions was performed to identify possible determinants of decreased lipid accumulation in siLmna-treated MSCs. ANOVA of the parameters for the multiple linear regression model showed that the nuclear perimeter as well as interactions between lamin A/C:area, lamin A/C:perimeter, lamin A/C:area:circularity, and lamin A/C circularity:perimeter comparisons were found to be significant (*p* < 0.05; Appendix A Appendix A). As shown in Appendix A Appendix A, a multiple linear regression model with these main affects, two-way interactions, and three-way interactions was able to predict the average lipid droplet intensity per cell with an R^2^ of 0.507, suggesting that decreased lamin A/C levels alone do not explain the decreased lipid droplet accumulation in lamin-A/C-depleted cells. Our RNA-seq analysis of the RNA samples taken from the same samples as the Western blotting and imaging data indicate that MSCs display an undifferentiated phenotype upon lamin A/C depletion, reflected by reduced expression of genes associated with adipogenic and lipid-related metabolic pathways (Figure 5c and Figure 6c). LIV treatment did not have a significant impact on adipogenic gene expression, indicating that lamin A/C and not low-intensity vibration has a greater influence on adipogenic differentiation.

In contrast to large shifts in transcription found after lamin A/C depletion, RNA-seq data from the same samples as the Western blotting data from Figure 4b indicated that only 21 genes for siCntl- and 74 genes for siLmna-treated cells were differentially expressed as a result of LIV treatment. Despite the lack of changes at the transcriptional level, lamin-A/C-depleted MSCs retain the ability to respond to mechanical signals and exhibit decelerated adipogenesis reflected by reduced adiponectin and lipid droplet intensity in LIV-treated cells. Although mechanical stimulation using LIV does not produce widespread alteration in mRNA expression, we did observe a distinct LIV-dependent signature characteristic of interferon-responsive genes. Relationships between the type 1 interferon signaling pathway and mechanical forces, specifically low-intensity forces such as shear strain and vibration, have been shown to inactivate interferons [44]. Regulating activation of the interferon signaling pathway is likely to impact the cellular interpretation and response to their physical environment, which may be done through controlling protein translation. Indeed, the absence of major transcriptome changes during adipogenesis in response to LIV points to post-transcriptional or post-translational regulatory events. However, further research is needed to elucidate any post-transcriptional or post-translational mechanoregulation effects of LIV mechanical stimulation on adipogenesis.

In this study, we did not validate our RNA-seq data using qPCR. While this may be seen as a limitation of the study, qPCR can be highly biased as it depends upon primer sequence design, variances in the target annealing temperatures, and the fact that results are usually referenced to a reference gene [45,46]. RNA-seq, in contrast, minimizes these inherent biases, especially the ones related to unforeseen variances in the reference gene due to experimental conditions, as the raw mRNA fragments are sequenced and aligned to the entire genome, thus providing an unbiased view of gene expression across all samples. Given the extremely close correlations between qPCR and RNA-seq as, reported by others [47,48,49,50,51], and the fact that we sequenced three independent biological replicates and ensured sequencing and alignment quality, we are confident that the mRNA data are representative of a true biologic response.

In summary, we showed that lamin A/C depletion results in decreased nuclear integrity, with more robust focal adhesions contributing to reduced adipogenic differentiation. This resulting adaptive response in the cytoskeletal structure maintains LIV-induced focal adhesion signaling and maintains the ability of LIV to limit adipogenesis of MSCs. Findings of this study indicate that lamin A/C is required for proper adipogenic commitment of MSCs and that the mechanical regulation of adipogenesis uses lamin-A/C-independent pathways to elicit a response in MSCs.

## 4. Materials and Methods

### 4.1. MSC Isolation

Bone-marrow-derived MSCs (mdMSC) from 8–10-week-old male C57BL/6 mice were isolated, as described in [52], from multiple mouse donors and MSC pools, providing a heterogenous MSC cell line. Briefly, tibial and femoral marrow was collected in RPMI-1640, 9% FBS, 9% HS, 100 µg/mL of penicillin/streptomycin, and 12 µM L-glutamine. After 24 h, non-adherent cells were removed by washing with phosphate-buffered saline and adherent cells cultured for 4 weeks. Passage 1 cells were collected after incubation with 0.25% trypsin/1 mM EDTA × 2 min, and re-plated in a single 175 cm^2^ flask. After 1–2 weeks, passage 2 cells were re-plated at 50 cells/cm^2^ in expansion medium (Iscove modified Dulbecco’s, 9% FBS, 9% HS, antibiotics, L-glutamine). mdMSCs were re-plated every 1–2 weeks for two consecutive passages up to passage 5 and tested for osteogenic and adipogenic potential and subsequently frozen.

### 4.2. Cell Culture, Pharmacological Reagents, and Antibodies

Fetal calf serum (FCS) was obtained from Atlanta Biologicals (Atlanta, GA, USA). Culture media, trypsin-EDTA, antibiotics, and phalloidin–Alexa Fluor 488 were obtained from Invitrogen (Carlsbad, CA, USA). MSCs were maintained in IMDM with FBS (10%, *v*/*v*) and penicillin/streptomycin (100 µg/mL). For phosphorylation measurements, the seeding cell density was 10,000 cells/cm^2^. For immunostaining experiments, the seeding cell density was 3000 cells/cm^2^. For phosphorylation measurements and immunostaining experiments, all groups were cultured for 48 h before beginning experiments and were serum-starved overnight in serum-free medium.

For adipogenic differentiation experiments, the seeding cell density was 10,000 cells/cm^2^. Cells were transfected 24 h after cell seeding with siRNA targeting *Lmna* (siLmna) or a control sequence (siCntl) using RNAiMax from Invitrogen. Adipogenic media and LIV treatment followed a previously published protocol, where 24 h after transfection, adipogenic media was added that contained dexamethasone (0.1 µM) and insulin (5 µg/mL) [19]. Cell cultures were incubated with the combined transfection media and adipogenic differentiation media for 7 days after adipogenic media was added with or without LIV treatment (2 × 20 min per day separated by 2 h).

The following antibodies were purchased: Akt (#4685), p-Akt Ser473 (#4058L), β-tubulin (D3U1W), and p-FAK Tyr397 (#328 3) from Cell Signaling Technology (Danvers, MA, USA); adiponectin (PA1-054) from Thermo Fisher Scientific (Rockford, IL, USA); and FAK (sc-558) and lamin A/C (sc-7292) from Santa Cruz Biotechnology (Dallas, TX, USA).

### 4.3. LIV and Strain

Vibrations were applied at peak magnitudes of 0.7 g at 90 Hz twice for 20 min separated by a 2 h rest period at room temperature. Uniform 2% biaxial strain was delivered at 10 cycles per minute for 20 min using the Flexcell FX-5000 system (Flexcell International, Hillsborough, NC, USA). Controls were sham-handled. During adipogenesis experiments, LIV was applied 24 h after initial transfection, a regimen we have previously shown be effective [26].

### 4.4. Isolation of Focal Adhesions

Cells were incubated with triethanolamine (TEA)-containing low-ionic-strength buffer (2.5 mM TEA, pH 7.0) for 3 min at RT and 1× PBS containing protease/phosphatase inhibitors. A Waterpik nozzle (Fort Collins, CO, USA; www.waterpik.com, accessed on 17 May 2021) held 0.5 cm from the plate surface at approximately 90° supplied the hydrodynamic force to flush away cell bodies, membrane-bound organelles, nuclei, cytoskeleton, and soluble cytoplasmic materials so that residual focal adhesions could be isolated, as reported previously [26].

### 4.5. siRNA-Silencing Sequences

For transient silencing of MSCs, cells were transfected with gene-specific small interfering RNA (siRNA) or control siRNA (20 nM) using RNAiMax (Thermo Fisher Scientific, Waltham, MA, USA) according to the manufacturer’s instructions. The following Stealth Select siRNAs (Invitrogen, Waltham, MA, USA) were used in this study: *Lmna* 5′-UGGGAGAGGCUAAGAAGCAGCUUCA-3′ and negative control for *Lmna* 5′-UGGGAGUCGGAAGAAGACUCGAUCA-3′.

### 4.6. Isolation of Nuclei for Young’s Modulus

MSCs were plated at 10,000 cells/cm^2^ cell density. For mechanical and structural testing, nuclei were isolated by scraping cells in PBS and then suspending cells in hypotonic solution, followed by centrifugation at 3000× *g*. Nuclei were then extracted by using percol (81% percol, 19% hypotonic buffer) and centrifugation at 10,000× *g*. Nuclei were then diluted in PBS and plated. Young’s modulus of the nuclei was determined using atomic force microscopy (AFM). For strain experiments, cells were plated on BioFlex collagen-I-coated silicone plates.

### 4.7. RNA-Seq

Seven days after adipogenic and LIV treatment, following the above protocols, three samples per group were sent to Novogene for RNA extraction and sequencing. Quality control of raw data was done using FASTQC. Read alignment of the genome to the raw reads was done using STAR [53]. Read count generation was performed using feature counts, and differential gene expression analysis was performed using DESEQ2 [54]. For analysis using fragments per kilobase of transcript per million mapped reads (FPKM), data were assessed as previously described [55,56,57,58]. Briefly, RNA-seq data were analyzed using a Mayo Bioinformatics Core called MAPRSeq v.1.2.1 [59], which includes TopHat 2.0.6 alignment [60] and gene expression quantification using HTSeq software [61]. Normalized gene counts were obtained from MAPRSeq as FPKM. Hierarchical clustering and principal component analysis were assessed and visualized using ClustVis [62]. Genes with significant differential expression compared to non-LIV controls (*p* < 0.05, log_2_FC > 1.0) as detected by DESEQ2 were analyzed via STRING [34]. Biological and cellular gene clusters that had a false discovery rate (FDR) less than 0.05 were further subjected to supervised analysis to determine proper naming of clusters. Sequencing data were deposited in the Gene Expression Omnibus of the National Institute for Biotechnology Information (GSE157056).

### 4.8. Immunofluorescence

Twenty-four hours after siRNA treatment against lamin A/C protein, cells were fixed with 4% paraformaldehyde. Cells were permeabilized by incubation with 0.3% Triton X-100. Cells were incubated in a blocking serum in PBS with 5% donkey serum (017-000-121, Jackson Immuno Research Laboratories, West Grove, PA, USA). Primary antibody solutions were incubated on the cells for 1 h at 37 °C, followed by secondary antibody incubation of either Alexa Flour 594 goat anti-rabbit (Invitrogen, Waltham, MA, USA) or Alexa Fluor 647 donkey anti-mouse. For nuclear staining, cells were incubated with NucBlue Hoechst stain (Thermo Fisher Scientific, Waltham, MA, USA). For actin staining, cells were incubated with Alexa Fluor 488–phalloidin (Life Technologies, Carlsbad, CA, USA). Primary and secondary concentrations were both 1:300.

### 4.9. Lipid Droplet Analysis

Seven days after adipogenic media and LIV treatment, cells were fixed and were stained with Oil Red O (Poly Scientific, Baywood, NY, USA, #k043), Lipid Spot 610 (Biotium, Fremont, CA, USA, #70069), and NucBlue Hoechst stain. Images were taken using a 20× objective and exported to quantify lipid droplet formation via a custom-made MATLAB program (MathWorks, Natick, MA, USA) previously published [63]. A minimum pixel intensity of 80 was used to isolate lipid droplet staining. The mean lipid droplet intensity per cell was calculated by dividing the sum of lipid droplet stain intensity by the nuclei count per image. For determining the effects of siLmna on lipid droplet formation, nuclear area, nuclear perimeter, and nuclear circularity, siCntl- and siLmna-treated cells were differentiated with previously stated adipogenic media and indomethacin (1 µg/mL) for 5 days. Cells were then stained for lamin A/C, as previously described, Lipid Spot 610, and NucBlue Hoeschst. Exported images were used to quantify lipid droplet formation, lamin A/C, nuclear area, nuclear perimeter, and nuclear circularity via the custom-made MATLAB program. A minimum pixel intensity of 80 was used to isolate lipid droplet staining and lamin A/C intensity.

### 4.10. Nuclear Morphology

To test the nuclear morphology to correlate with the mechanical constraint on the nucleus, as measured by AFM, MSCs were seeded at 3000 cells/cm^2^ on plastic slide chambers (iBIDI µslide # 80421). Then, 72 h after siRNA treatment against lamin A/C protein, cells underwent immunofluorescence staining following the above protocols for lamin A/C and DNA (Hoechst 33342; Life Technologies, Carlsbad, CA, USA) and/or were stained against actin (Alexa Fluor 488–phalloidin; Life Technologies, Carlsbad, CA, USA). Z-stack confocal 3D images were obtained with Zeiss LSM 710 with a separation interval of 0.15 µm. Z-stack images were analyzed using IMARIS software.

### 4.11. Western Blotting

Whole-cell lysates were prepared using radio immunoprecipitation assay (RIPA) lysis buffer (150 mM NaCl, 50 mM Tris HCl, 1 mM EDTA, 0.24% sodium deoxycholate, 1% Igepal, pH 7.5). To protect the samples from protein degradation, NaF (25 mM), Na_3_VO_4_ (2 mM), aprotinin, leupeptin, pepstatin, and phenylmethylsulfonylfluoride (PMSF) were added to the lysis buffer. Western protein amounts were normalized to 15 µg through BCA protein assay (Thermo Fisher Scientific, Waltham, MA, USA; #23225). Cell fractionation protein amounts were normalized to 10 µg via detergent-compatible Bradford assay (Thermo Fisher Scientific, Waltham, MA, USA; #23246). LDHA and PARP were used as fractionation total loading protein controls for nuclear and cytoplasmic fractions, respectively. Whole-cell lysates (15 µg) were separated on 10% polyacrylamide gels and transferred to polyvinylidene difluoride (PVDF) membranes. Membranes were blocked with milk (5%, *w*/*v*) diluted in Tris-buffered saline containing Tween 20 (TBS-T, 0.05%). Blots were then incubated overnight at 4 °C with appropriate primary antibodies. Following primary antibody incubation, blots were washed and incubated with horseradish-peroxidase-conjugated secondary antibody diluted at 1:5000 (Cell Signaling Technology) at RT for 1 h in 5% milk in TBST-T. Chemiluminescence was detected with ECL plus (Amersham Biosciences, Piscataway, NJ, USA). At least three separate experiments were used for densitometry analyses of Western blots, and densitometry was performed using NIH ImageJ software 1.52.

### 4.12. Statistical Analysis and Reproducibility

Results for densitometry were presented as the mean ± SEM. Densitometry and other analyses were performed on at least three separate experiments. Differences between groups were identified by two-tailed Student’s *t*-test. Analysis of nuclear morphology and Young’s modulus measurement were performed using the Whitney–Mann test, and results were presented as the mean ± STD. Differential gene expression analysis via DESEQ2 was done using Wald test. Image analysis groups were analyzed using the Kruskal–Wallis test. *p*-Values of less than 0.05 were considered significant. All experiments were conducted in triplicate to ensure reproducibility.

## Figures and Tables

**Figure 1 ijms-22-06580-f001:**
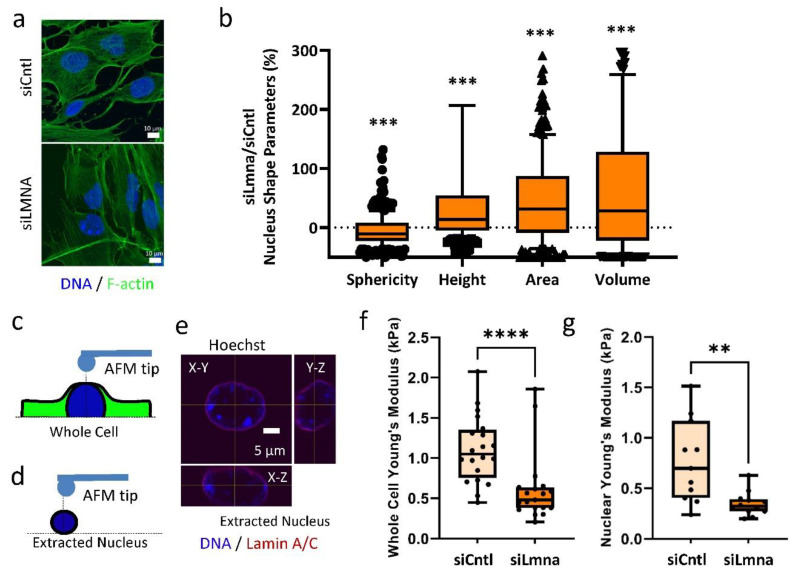
siRNA depletion of lamin A/C weakens the nuclear elastic modulus in MSCs: (**a**) Confocal Image of F-actin (phalloidin, green) and nucleus (Hoechst, blue). Scale bar: 10 µm. (**b**) Nuclear sphericity decreased by 8% in MSCs treated with Lamin A/C specific siRNA (siLmna) compared to MSCs treated with a non-specific control siRNA (siCntl) (*p* < 0.05, *n* = 342). Nuclear area of siLmna treated cells showed a 32% increase when compared to siCntl (*p* < 0.05, *n* = 342). Nuclear volume siLmna treated cells increased by 31% compared to siCntl (*p* < 0.05, *n* = 342). When compared to the nuclear height of siCntl MSCs, siLmna treated cells had increased nuclear height of 12% (*p* < 0.05, *n* = 342). (**c**) Schematic of AFM probe tip testing whole cell Young’s modulus in live MSCs. (**d**) Depiction of AFM probe tip testing live extracted nucleus. (**e**) Confocal image of extracted nucleus depicting its orthogonal views from X-Y, X-Z, Y-Z planes (Hoechst, blue; Lamin A/C, Red) Scale bar: 5 µm. (**f**) Whole cell Young’s modulus of the siLmna group was 45% lower when compared to the siCntl group (*p* < 0.0001, *n* = 16). (**g**) Young’s modulus of extracted live nucleus in siLmna MSCs remained 55% lower when compared to siCntl MSCs (*p* < 0.01, *n* = 13). Results are presented as mean ± STD. Group comparisons were made via non-parametric Mann Whitney tests. *p* < 0.05, ** *p* < 0.01, *** *p* < 0.001, **** *p* < 0.0001 against control.

**Figure 2 ijms-22-06580-f002:**
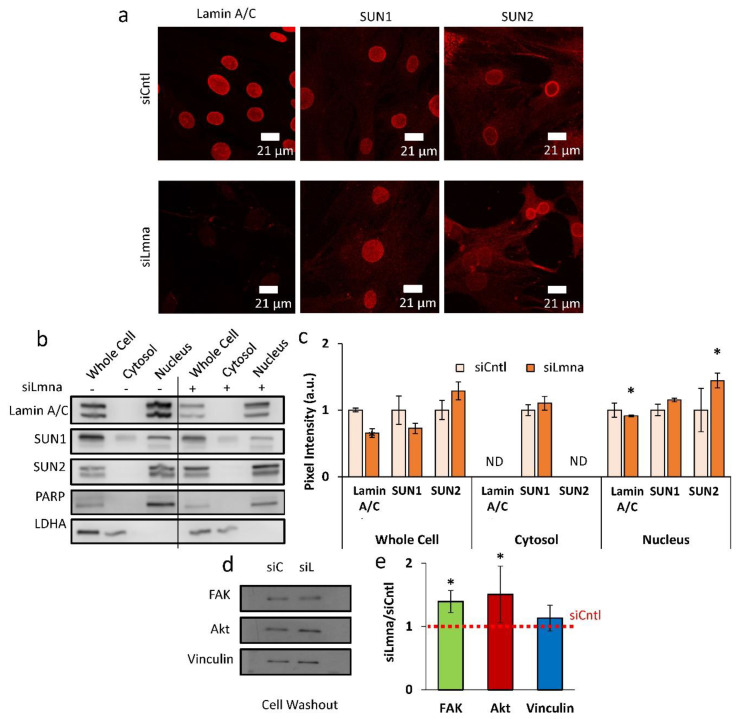
siRNA depletion of lamin A/C increases SUN2 nuclear levels and focal adhesion proteins: (**a**) Confocal images of cells treated with the siCntl and siLmna siRNA groups. Primary antibodies targeted lamin A/C, SUN1, and SUN2. Hoechst stained images can be found in Appendix A. (**b**) Representative western blots of cell fractionations (whole cell, cytosol and nucleus) with cells treated with either siCntl or siLmna. Primary antibodies targeted lamin A/C, SUN1, SUN2, PARP, and LDHA. Line represents removal of protein ladder marker lane, uncropped blots are provided in Appendix A. (**c**) Analysis of western of cell fractionation western blots (*n* = 3/grp). siLmna treated cells had a 44% increase in nucleus fraction (*p* < 0.05) compared to siCntl samples. SUN1 showed no detectable changes. ND represents non-detectable levels. (**d**) Representative western blot of focal adhesion proteins following a cell washout. Primary antibodies targeted of FAK, Akt, and Vinculin in siCntl and siLmna siRNA treated cells. (**e**) Densitometry analysis showed that, when compared to siCntl levels siLmna treated MSCs showed increased levels of total FAK (39%, *p* < 0.05) and total Akt (50%, *p* < 0.05), no change in Vinculin was detected (*n* = 3/grp). Results are presented as mean ± STE. Scale bar: 21 µm. Group comparisons were made via parametric two-tailed Student *t*-test (C) or one-way ANOVA followed by a Newman-Keuls post-hoc test. * *p* < 0.05 against control.

**Figure 3 ijms-22-06580-f003:**
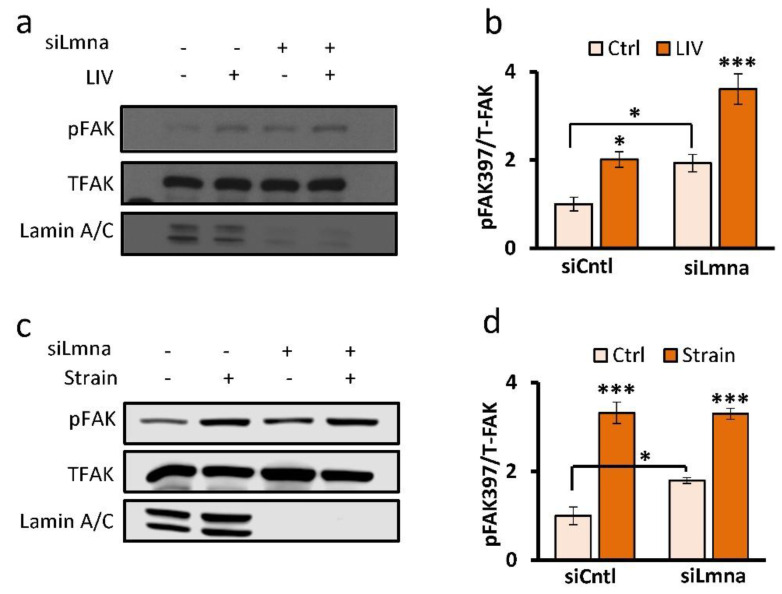
Focal adhesions maintain response to mechanical stimulus in lamin A/C depleted MSCs: (**a**) Representative western blots for pFAK (Tyr 397), TFAK, and lamin A/C in siCntl and siLmna treated cells groups treated with 2 bouts of LIV (20 min, 90 Hz, 0.7 g) separated by 2 hour rest period. LIV treated samples had a 2-fold increase of pFAK compared to non-LIV. (**b**) Analysis of western image of pFAK, TFAK, and lamin A/C during LIV (*n* = 4/grp). The non-LIV siLlmna group had a 92% (*p* < 0.05) increased basal pFAK compared to the non-LIV siCntl group. In response to LIV, both siCtnl and siLmna treated MSCs elicited 101% (*p* < 0.05) and 87% (*p* < 0.001) increases in pFAK, respectively, compared to non-LIV controls. (**c**) Representative western blots for pFAK (Tyr 397), TFAK, and Lamin A/C of the siCntl and siLmna groups treated with a single bout strain (20 min, 0.1 Hz, 2% strain). (**d**) Analysis of pFAK, TFAK, and lamin A/C immediately after strain application (*n* = 4/grp). The non-strain siLmna group had a 79% (*p* < 0.05) increased basal pFAK compared to the non-strain siCntl group. In response to strain, pFAK levels were elevated by 331% (*p* < 0.001) and 83% (*p* < 0.001) in siCtnl and siLmna treated MSCs respectively. Results are presented as mean ± STE. Group comparisons were made via one-way ANOVA followed by a Newman-Keuls post-hoc test. * *p* < 0.05, *** *p* < 0.001, against control or against each other.

**Figure 4 ijms-22-06580-f004:**
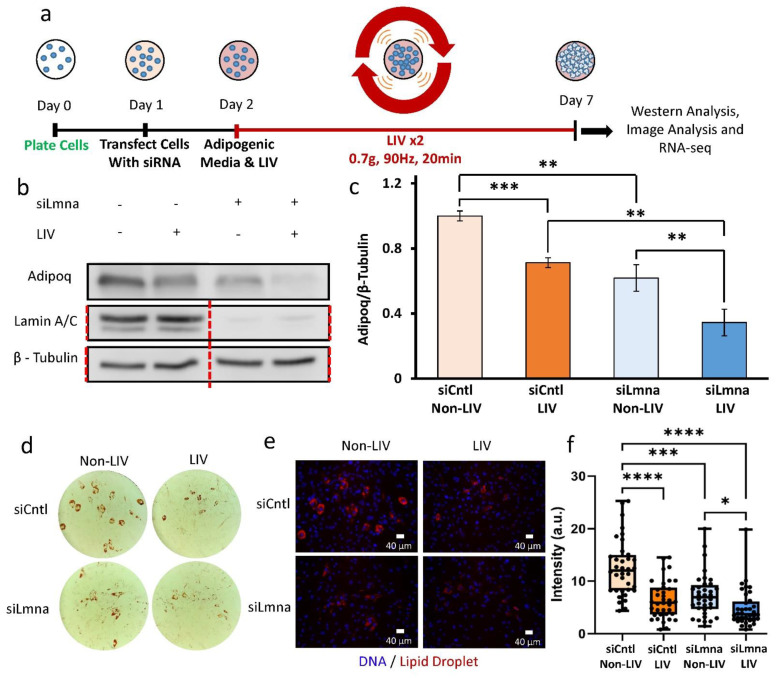
Application of daily LIV treatment decreases adipogenic differentiation in MSCs: (**a**) On day 0 cells were plated. Then, on day 1 cells were transfected with siRNA. On day 2 adipogenic media was placed on cells and cells were treated with LIV for 20 min, twice daily. Once cells differentiated, cells were pulled off for western analysis, image analysis, and RNA-seq. (**b**) Representative western blots of cells treated with siCntl and siLmna after 7 days of adipogenic induction with and without LIV treatment. Adiponectin (Adipoq), lamin A/C, and β-Tubulin were targeted. Lamin A/C and β-Tubulin were imaged on the same plot. Red line represents western blot cropped for alignment; uncropped blots were provided in Appendix A. (**c**) Relative levels of adiponectin of the siCntl and the siLmna groups. Adipoq levels in iLmna treated MSCs with no LIV were decreased by 39% (*p* < 0.01, *n* = 4) compared to siCntl MSCs with no LIV. LIV treated samples had 30% reduction in Adipoq levels compared to non-LIV controls for siCntl treated cells (*p* < 0.001, *n* = 3/grp). siLmna treated cells treated with LIV had a reduction of Adipoq levels of 44% compared to non-LIV samples (*p* < 0.01, *n* = 3/grp). siLmna cells treated with LIV compared to siCntl cells with LIV treatment had a 51% reduction in Adipoq (*p* < 0.01, *n* = 3/grp). (**d**) Representative Oil-Red-O images. (**e**) Representative images of lipid droplet staining corresponding to Oil-Red-O images Scale bars: 40 µm. (**f**) Analysis of the mean lipid droplet intensity per cell. LIV treated siCntl cells experienced a decrease of lipid droplet mean intensity by 49% (*n* = 36, *p* < 0.0001). siLmna cells treated with LIV had a decrease of 44% compared to non-LIV siLmna samples. (*n* = 36, *p* < 0.05). siLmna non-LIV samples had 42% less mean intensity compared to siCntl non-LIV samples (*n* = 36, *p* < 0.001). Western results are presented as mean ± STE. Western group comparisons were made via parametric two-tailed Student *t*-test. Lipid droplet group comparisons were made with Kruskal-Wallis Test. * *p* < 0.05, ** *p* < 0.01, *** *p* < 0.001, **** *p* < 0.0001, against control.

**Figure 5 ijms-22-06580-f005:**
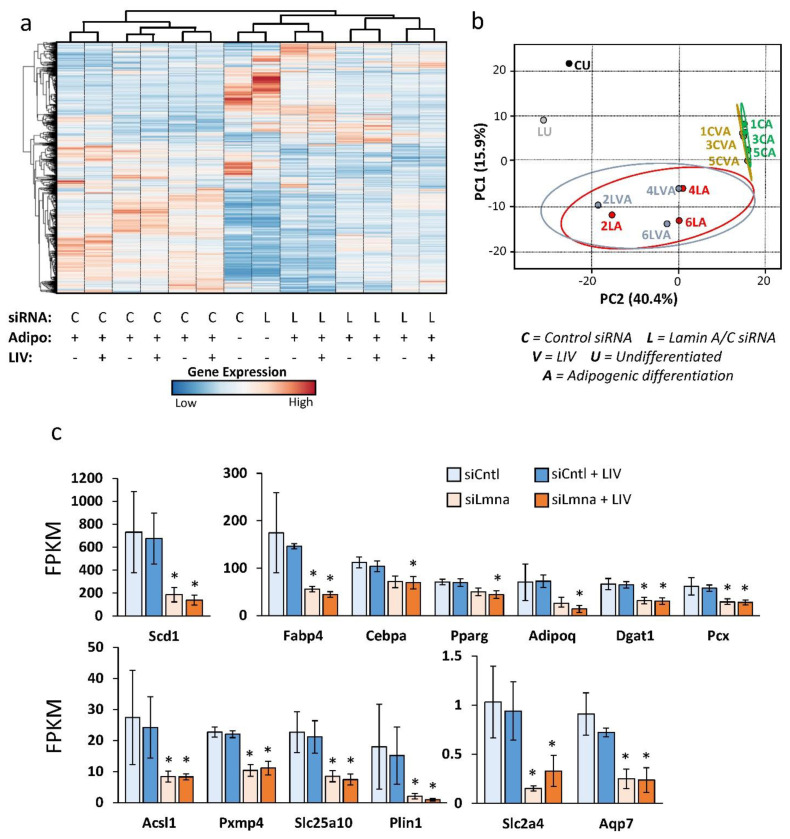
Differential effect of lamin A/C depletion and LIV on mRNA transcription during adipogenic differentiation: (**a**) Heat map of genes with average minimal expression of 0.3 FPKM, *t*-test *p* < 0.05, and fold change greater than 1.4. Unit variance scaling is applied to rows. (**b**) Principle component plot where principal component 1 and principal component 2 that explain 40.4% and 15.9% of the total variance, respectively. Prediction ellipses are such that with probability 0.95, a new observation from the same group will fall inside the ellipse. *n* = 14 data points. (**c**) Average FPKM values of DESEQ2 analysis for differentially expressed genes related to adipogenic phenotype. Results are presented as mean ± STE. * *p* < 0.05 and fold change > 1.0 compared to siCntl.

**Figure 6 ijms-22-06580-f006:**
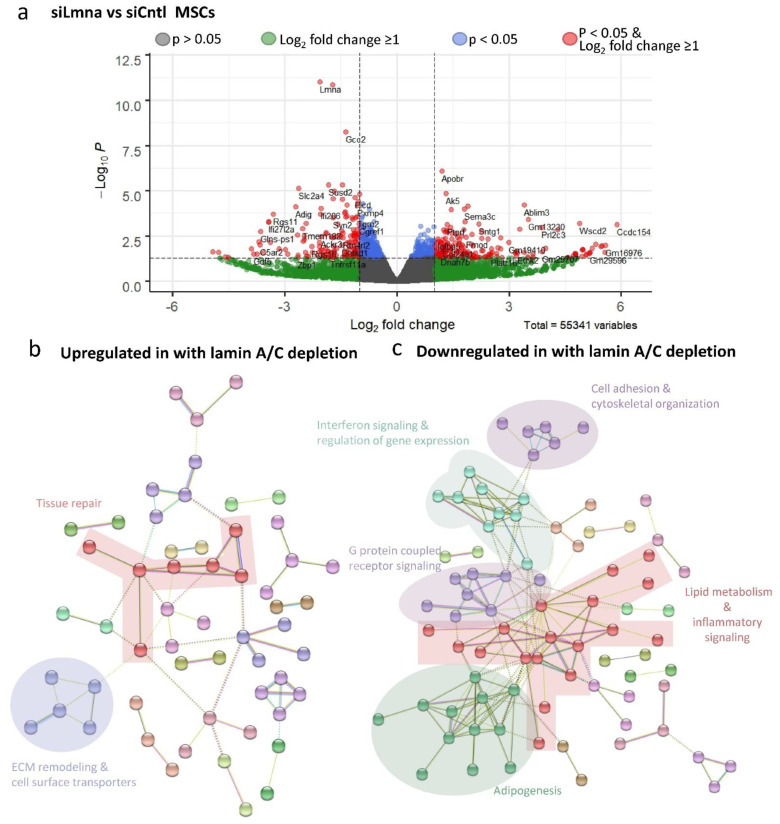
Lamin A/C depletion impedes adipogenic transcription in MSCs: (**a**) Volcano plot of siLmna compared to siCntl under adipogenic conditions. Genes with Wald values greater than *p* value of 0.05 are colored in grey. Genes with differential gene expression equal to or larger than 2-fold ($Log_{2}$ = 1) but have Wald values greater than *p* value of 0.05 are colored in green. Genes colored with blue have Wald values smaller than *p* value of 0.05, but differential gene expression less than 2-fold ($Log_{2}$ = 1). Genes with Wald values smaller than *p* value of 0.05 and differential gene expression equal to or larger than 2-fold ($Log_{2}$ = 1) are colored in red. Grouping of five or more associated genes were highlighted and subsequently subjected to a supervised analysis of biologic function. (**b**) Upregulated genes were associated with cellular processes included tissue repair, ECM remodeling and cell surface transporters. (**c**) Downregulated gene groups included, cell adhesion and cytoskeletal organization, interferon signaling and regulation of gene expression, G-protein coupled receptor signaling, lipid metabolism and paracrine inflammatory signaling and adipogenic phenotype.

**Figure 7 ijms-22-06580-f007:**
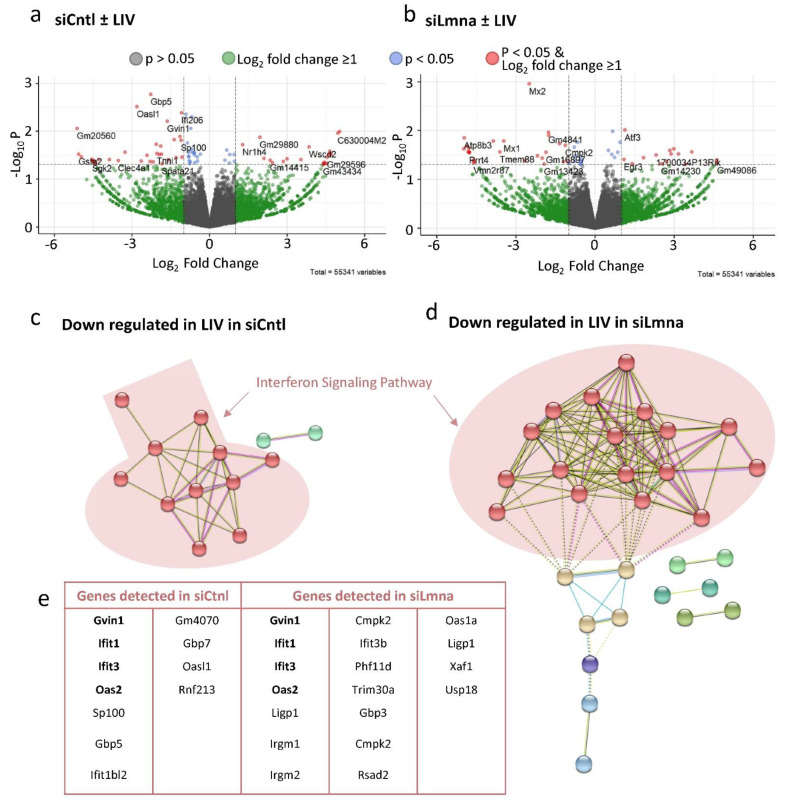
Low Intensity Vibration Decreases Interferon Signaling Pathway Gene Expression: (**a**) Volcano plot comparing the siCntl adipogenesis with or without LIV treatment (siCntl ± LIV). (**b**) Volcano plot comparing the siLmna adipogenesis with or without LIV treatment (siLmna ± LIV). Genes with Wald values greater than *p* value of 0.05 are colored in grey. Genes with differential gene expression equal to or larger than 2-fold ($Log_{2}$ = 1) but have Wald values greater than *p* value of 0.05 are colored in green. Genes colored with blue have Wald values smaller than *p* value of 0.05, but differential gene expression less than 2-fold ($Log_{2}$ = 1). Genes with Wald values smaller than *p* value of 0.05 and differential gene expression equal to or larger than 2-fold ($Log_{2}$ = 1) are colored in red. Both siCtnl (**c**) and siLmna (**d**) showed downregulation of genes closely associated with interferon signaling pathway. (**e**) Cells treated with siCntl had 11 genes associated with interferon signaling pathway while siLmna treated cells had 16 genes associated with the interferon signaling pathway that had decreased expression. **Bolded** gene names (*Gvin*, *Ifit1*, *Ifit3*, and *Oas2*) names were found in both siCntl and siLmna treated samples.

## Data Availability

The datasets generated and/or analyzed during the current study are available from the corresponding author on reasonable request.

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
