# Peer review of "Lamin A/C Is Dispensable to Mechanical Repression of Adipogenesis"

_ijms, 2021, doi:10.3390/ijms22126580_

Round 1

Reviewer 1 Report

Major comments: 1. In Figure 2a, DAPI images of the same cells should be added to verify that there are cells in siLmna image. 2. Fig. 2c: the figure does not support the conclusion that "siLmna treated cells had 29% increase of sun-2 in whole cells, 122% in cytoplasm, and 44% in nucleus fraction" Same for sun-1 3. Figure 5 and later figures: at least some of the findings should be validated using RT-qPCR 4. The section 2.7 and lines 196-197 contradict each other

Minor comments: 1. Throughout the manuscript the notation regarding sun-1 and sun-2 is incorrect with regards to gene/protein names 2. Line 184 should read "minimal levels of 0.3 FKPM" 3. There is a lot of jargon which needs to be removed, e.g. lines 246-248, the title of Fig. 7 4. Misspellings in Fisher Scientific, Hoechst stain 5. the "Funding" and "Institutional Review Board Statement" sections are not completed

Reviewer 2 Report

In this manuscript by Goelzer et al, the authors investigated the impact of Lamin A/C depletion and mechanical signals such as low intensity vibration (LIV) on adipogenic differentiation of mesenchymal stem cells (MSCs). While both Lamin A/C down-regulation and LIV application inhibit adipogenic differentiation with concomitant activation of focal adhesions, the downstream mechanisms may be independent. This is a nice study connecting mechanical force, cytoskeletal communication and nuclear structure with adipogenic differentiation. However, there are a few concerns that need to be addressed before this manuscript is in a publishable fashion. Specific comments are as follows:

1. The major concern of this study is that the RNA-Seq results in Fig 5 did not reflect what was observed in immunoblotting and lipid staining (Fig 4): LIV treatments did not affect expression of adipogenic genes. The authors discussed that the discrepancy may be due to regulation of LIV at a post-translational level, e.g. activation of the beta-catenin pathway. However, this statement has to be experimentally verified, as beta-catenin activation does affect mRNA expression of adipogenic genes in various other cases.

2. In Fig 2b, 2c, line 130, the authors stated, "All the measurements were normalized to whole cell siCntl protein amounts…". I am not sure whether this is appropriate. I may assume that siRNA transfection does not affect total protein levels, but I cannot assume that every sample has the same amount of protein as individual variations can occur by hand.

3. Also in Fig 2b, are PARP and LDHA nuclear and cytoplasmic markers? They do not look equally loaded in siCntl and siLmna samples so this goes back to the question in 2).

4. In Figure 2d and 2e, the differences in band intensity do not look as dramatic in number as in the real blots. By the way, how are the proteins measured normalized to ensure equal loading?

5. In Fig 3, strain has a different effect on FAK phosphorylation compared with LIV. This deserves discussion.

Round 2

Reviewer 1 Report

The authors have addressed my concerns sufficiently.

Reviewer 2 Report

The authors have addressed all the questions. Despsite that how LIV administration molecularly affects adipogenic differentiation remains an open question, the authors have provided sufficient data to delineate their observations.